# Visceral Leishmaniasis: Epidemiology, Diagnosis, and Treatment Regimens in Different Geographical Areas with a Focus on Pediatrics

**DOI:** 10.3390/microorganisms10101887

**Published:** 2022-09-21

**Authors:** Sara Scarpini, Arianna Dondi, Camilla Totaro, Carlotta Biagi, Fraia Melchionda, Daniele Zama, Luca Pierantoni, Monia Gennari, Cinzia Campagna, Arcangelo Prete, Marcello Lanari

**Affiliations:** 1Specialty School of Pediatrics, Alma Mater Studiorum, University of Bologna, 40126 Bologna, Italy; 2Pediatric Emergency Unit, IRCCS Azienda Ospedaliero-Universitaria di Bologna, 40138 Bologna, Italy; 3Pediatric Oncology and Hematology “Lalla Seràgnoli”, IRCCS Azienda Ospedaliero-Universitaria di Bologna, 40138 Bologna, Italy

**Keywords:** visceral leishmaniasis, tropical diseases, neglected diseases, Leishmania, protozoa, pediatrics

## Abstract

Visceral Leishmaniasis (VL) is a vector-borne disease caused by an intracellular protozoa of the genus *Leishmania* that can be lethal if not treated. VL is caused by *Leishmania donovani* in Asia and in Eastern Africa, where the pathogens’ reservoir is represented by humans, and by *Leishmania infantum* in Latin America and in the Mediterranean area, where VL is a zoonotic disease and dog is the main reservoir. A part of the infected individuals become symptomatic, with irregular fever, splenomegaly, anemia or pancytopenia, and weakness, whereas others are asymptomatic. VL treatment has made progress in the last decades with the use of new drugs such as liposomal amphotericin B, and with new therapeutic regimens including monotherapy or a combination of drugs, aiming at shorter treatment duration and avoiding the development of resistance. However, the same treatment protocol may not be effective all over the world, due to differences in the infecting *Leishmania* species, so depending on the geographical area. This narrative review presents a comprehensive description of the clinical picture of VL, especially in children, the diagnostic approach, and some insight into the most used pharmacological therapies available worldwide.

## 1. Epidemiology, Microbiology, and Transmission

Visceral leishmaniasis (VL) is a potentially fatal vector-borne disease reaching the second and seventh place among tropical diseases in mortality and loss of disability-adjusted life years, respectively [1,2]. However, being mainly considered as a disease of developing countries, the pharmaceutical industry shows little interest in investing in its research, so VL is included among “the neglected diseases” [3,4]. In addition, in the most affected areas, access to health care can be difficult and drugs are often not affordable, making the death rate even higher [5].

According to the World Health Organization (WHO), almost 13,000 cases of VL occurred in 2020 [6]. VL is endemic in over 70 countries spread across all continents, except Antarctica and Australia, with an estimated at-risk population of 200 million people [7]. However, VL is mostly distributed in seven countries, i.e., Brazil, Ethiopia, India, Kenya, Somalia, South Sudan, and Sudan, where more than 90% of the worldwide VL cases are reported [7]. An estimated 500,000 new cases of VL and 50,000 deaths occur annually, which are thought to be underestimated [5,8].

The observed trend of VL infection around the world increased between 1998 and 2005, was globally stable between 2005 and 2007, and notably declined between 2007 and 2009, consistently with a reduction in the South-East Asia region. In 2011 a global peak was observed and was likely caused by an African hotspot with outbreaks in South Sudan and Kenya, and by rising trends in the Indian regions. After 2011, VL cases in the South-East Asia Region and therefore globally drastically decreased [9]. Comparing the three main endemic areas, the number of cases in Eastern Africa increased from 2015 to 2016, the Indian subcontinent showed a decreasing trend in the same two-year period, while VL cases in Brazil remained stable in 2015 and 2016 [9].

The Eastern Mediterranean and European regions are also considered endemic areas, although the reported cases are fewer compared to South-East Asia and Africa. Contrary to other exotic vector-borne diseases, namely Dengue and Chikungunya, which caused health emergencies in Europe in the last decade and thus raised the interest of the scientific community, VL has been neglected [10,11], leading to sub-optimal knowledge of healthcare professionals about its prevalence, symptoms, and diagnostic options. The European Center for Disease Prevention and Control (ECDC) conducted surveillance of VL cases in Europe from 2014 to 2019. Results reported an increased incidence of VL in Armenia, Azerbaijan, France, Greece, and Libya, and a significant decrease in Albania, Algeria, Croatia, Georgia, Morocco, and Tunisia. Moreover, ECDC observed emerging evidence of VL on a sub-national scale in endemic countries such as Italy, Greece, and Spain and former non-endemic countries such as Romania [12]. A few years ago, an outbreak of VL was observed in Northern Italy: 14 cases were reported in the Bologna city area from November 2012 to May 2013 [13]. Similarly, between 1 July 2009 and 31 December 2012, a community outbreak of leishmaniasis occurred in the southwestern region of Madrid, Spain [14]. The reported outbreak cases were 446, 35.9% of which were VL and whose number was fivefold higher compared to the whole cases of the previous years [14].

The term Leishmaniasis covers a wide range of clinical manifestations due to infections by protozoa of the *genus Leishmania* [15,16,17]. In particular, the genus includes the *subgenera Leishmania* and *Vianna* [15]. VL is mainly caused by *Leishmania donovani* and *Leishmania infantum*, the latter also called *Leishmania chagasi* in South America. VL due to *L. donovani* occurs in Southeast Asia, in particular India, Bangladesh, and Nepal, and in Eastern Africa, especially Sudan, Ethiopia, Kenya, and Somalia [3,7,8]. It can affect people of all ages, even if in endemic areas its incidence is higher among children due to the acquired immunity of adults [8]. In both regions, VL exhibits cyclical patterns of occurrence: in particular, the incidence increases over 2 to 5 years with a peak and then drops for some years [18]. In 2014, Sudan, South Sudan, and Ethiopia reported more than 14,000 cases, and about 10,000 were observed in India and Bangladesh [18]. 

*L. infantum* causes VL in the Mediterranean area, in the Middle East, Afghanistan, Iran, Pakistan, and Brazil. Rare cases have been observed in Central Asia and Latin and Central America excluding Brazil [19], where 90% of reported VL cases in the Americas occur [9]. Children aged less than 10 years and immunosuppressed individuals are more likely to manifest a clinical disease due to *L. infantum* than immunocompetent adults [8]. 

The vector responsible for *Leishmania* transmission is the female sandfly, which is a haematophagous, noiseless, 2–3 mm long arthropod whose color ranges from black to white [20]. Most sandflies are active outdoors from dusk to dawn, although some of them can also bite indoors and in the daylight [20]. For survival, the *Leishmania* parasite needs a mammalian reservoir. In the case of *L. donovani*, the reservoir is represented by infected humans, so that in the Asiatic and African regions transmission occurs anthroponotically (anthroponotic cycles). In the case of *L. infantum*, the reservoir is represented by dogs and, less frequently, hares, wild rabbits, and other wild mammals, so that, in the Mediterranean area and in Latin America, VL is a zoonosis (zoonotic cycles) [19,20,21,22,23]. The two major morphological forms of the protozoan are the extracellular promastigote within the sandfly and the intracellular amastigote in the monocyte-macrophage cells of the mammalian host [20]. The amastigotes are able to multiply inside the infected cells until they are released by cell lysis in order to infect other macrophages. These parasite forms are relevant to diagnostic purposes, in terms of direct visualization in different tissue specimens. The intracellular amastigotes seen on staining of clinical specimens are known as Leishman-Donovan bodies [15,24]. Figure 1 reports the life cycle of *L. infantum* and *L. donovani* and the main symptoms of VL. 

VL is a poverty-related disease affecting people in rural areas, although in some countries cases related to peri-urbanization have also been described. Outbreaks occur during massive migration of susceptible people to endemic regions or changing of the natural habitat of sandflies, such as deforestation [8]. In certain areas, such as Eastern Africa, the rising numbers of immunosuppressed people due to HIV result in an increase in VL cases [8].

## 2. Clinical Manifestations

Symptoms caused by this protozoa infection can be various and depend on the interaction between the host immune response and the infectious features of *Leishmania* species. Although the course of many *Leishmania* infections is asymptomatic, three clinical syndromes can be identified: cutaneous leishmaniasis, mucosal leishmaniasis, and VL [15,16,25]. The incubation period varies from a few weeks up to 6 months or sometimes years [8,26]. If untreated, VL can be fatal due to the disease itself or from infectious or hemorrhagic complications. On the other hand, VL may also behave asymptomatically or as a latent infection and become manifest after years in case of development of immunodeficiency of any etiology [15,25]. 

Following the sting of an infected sandfly, *Leishmania* spreads and replicates in the reticuloendothelial system. The typical clinical picture in children, as well as in adults, is the consequence of the spread of parasites in macrophages and their propagation to lymph nodes and lymphoid organs evading the immune system of the host [25,27]. In particular, children under 10 years of age have a higher risk of developing VL due to an immature innate immune response, lack of exposition to the pathogen and acquired immunity, and higher rates of malnutrition compared to adults [25]. 

Symptom onset is generally subacute, insidious, and slowly progressive, and rarely acute [15,16,25]. Signs and symptoms include fever, weight loss, splenomegaly, variable hepatomegaly, pancytopenia (more frequently anemia and thrombocytopenia), elevated liver enzymes, and hypoalbuminemia with hypergammaglobulinemia [13,15,25]. Pancytopenia varies in frequency and severity, but it is usually associated with prolonged illness [28]. Normochromic normocytic anemia is commonly found during illness among adults and children but seems to be more severe in children, as reported by Al-Jurrayan et al. who reviewed the cases of 94 children with VL and found that all of them were anemic [28,29]. Given that parasites reproduce not only in the spleen and liver, but also in bone marrow, severe pancytopenia can occur as a result of a combination of several factors: suppression of hematopoiesis, splenic sequestration, and hemolysis.

Fever is usually remittent with twice-daily spikes, but can be intermittent or less often continuous. Patients with VL may complain of abdominal fullness and discomfort due to the splenomegaly. The spleen may be firm and tender or cause pain when it enlarges rapidly. Patients can also present with hepatomegaly, which is usually less marked than splenomegaly [8]. A recent retrospective study including patients affected by VL aged less than 15 years observed that 45% of children had splenic nodules [27,30]. These patients had a median age lower than that of the group without splenic nodules (8 months vs. 21 months, respectively) [30]. 

Lymphadenopathy is observed mostly in East African VL (e.g., South Sudan), but it is uncommon in other endemic areas [15,26]. 

With the progression of the disease, hepatic dysfunction may develop with jaundice and ascites. Hepatic failure and thrombocytopenia can lead to hemorrhagic complications such as spontaneous bleeding from nasal and oral mucosa. In case of invasion of the intestine by protozoa, diarrhea, malabsorption, and hypoalbuminemia can also occur [25]. Hypergammaglobulinemia, predominantly IgG derived from B-cells activation, is quite common [15,26].

The term “kala-azar”, also known as “black sickness”, derives from the greyish discoloration of the skin developed by some patients in South Asia, probably a result of a cytokine-induced increase in the production of adrenocorticotropic hormone [25]. 

### Complications

The most severe, potentially fatal complications of VL are disseminated intravascular coagulation (DIC) [28,31,32,33] and hemophagocytic lymphohistiocytosis (HLH) [28,34,35,36,37]. Moreover, months to years after VL caused by *L. donovani* infection, post Kala-azar dermal leishmaniasis (PKDL) may appear [38]. Patients with VL may also develop a mild renal involvement [26,27,28,29]. 

DIC is rare: in the literature, only a few cases of DIC in VL-infected patients are described [28,33]. However, in countries where VL is endemic, patients presenting with DIC and pancytopenia should be tested for Kala-azar [28,32,33].

Another life-threatening complication is HLH, which is a disorder related to the uncontrolled activation of cytotoxic T-lymphocytes and natural killer (NK) cells [36,37]. Although HLH is a rare complication of VL, it has to be suspected because it increases mortality and may require specific therapies [36,37], but its recognition can be challenging because of the overlapping clinical manifestations [39]. Children with HLH present with a constellation of symptoms and signs that include fever, hepatosplenomegaly, neurologic dysfunction (such as encephalitis, seizures, or coma), dermatologic manifestations, and markers of liver dysfunction (such as jaundice or ecchymosis) [34]. In particular, children affected by HLH, compared with adults, have hepatomegaly in almost all cases (95% vs. 18–67%) and neurological manifestations associated with a poor prognosis (33% vs. 9–25%) [34]. Diagnosing HLH is challenging, especially in young children who present with high fever and hepatosplenomegaly. Gagnaire et al. conducted a retrospective study which included 12 cases of young children who had been diagnosed with familiar or infection-induced HLH instead of VL: the diagnostic delay was caused by the negativity of serologic tests for *Leishmania* at onset, lack of history of visiting foreign countries, and the very young age of children [40]. In a recent review, Rajagopala et al. found that another confounding element is the negativity of the first bone marrow aspiration in 64.7% for HLH and 36.3% for Leishman-Donovan bodies [37]. Moreover, despite HLH being thought to be a rare complication of VL, in a retrospective cohort of 127 children affected by VL in a Brazilian tertiary care hospital, this affection was observed in up to 28% of cases [36]. 

PKDL is a cutaneous complication of VL due to *L. donovani* in people who recovered from the disease, usually after treatment [8,41,42]. PKDL is characterized by a macular, maculopapular, and nodular rash which usually arises around the mouth and then spreads to other parts of the body such as the shoulders, trunk, and extremities [8,41,42]. The connection with treatment seems to have an immunologic pathogenesis: during VL, peripheral blood mononuclear cells do not produce interferon γ, but they start releasing it after therapy, causing skin inflammation and the resulting typical lesions. PKDL occurs in Sudan in 50–60% of cases and less frequently in India in 5–10% of cases and develops within 6 months and 2–3 years, respectively, after VL treatment. Interestingly, 8% of cases have no previous history of VL; others present with concomitant VL and PKDL or may develop PKDL while still on VL treatment (up to 18%) [42]. Patients in whose skin lesions parasites are found play a potential role in the transmission of the disease. Indian forms of PKDL usually need therapy, whereas PKDL in Sudan is self-healing in the majority of cases [8,41,42]. 

Renal impairment caused by leishmaniasis includes interstitial and glomerular dysfunction [43,44]. The mechanism of renal involvement in VL pathophysiology is not fully understood, but it seems to be the result of an immune complex disease, which is responsible for proliferative glomerulonephritis and interstitial nephritis, as in other parasitic infections [43]. Liborio et al. tried to investigate the incidence of acute kidney injury in children with VL and define its relative risk factors [43]. They conducted a retrospective cohort study which comprised 146 patients under 14 years and, according to the pediatric Risk, Injury, Failure, Loss, End-Stage Renal Disease criteria [45], they identified risk, injury, and failure in 67.2%, 31.3%, and 1.5%, respectively [43]. Children who developed acute kidney injury were younger and had jaundice; secondary infections, serum albumin decrement, and high serum globulin were found as risk factors. 

## 3. Diagnosis

Overall, the clinical diagnosis of VL is difficult because its presentation overlaps with other infections like typhoid fever, tuberculosis, brucellosis, malaria, or some hematologic malignancies [21]. Traditionally, in a child with clinical suspicion of VL (febrile splenomegaly, hepatomegaly, loss of weight, and laboratory signs such as pancytopenia and hypergammaglobulinemia or hemophagocytic syndrome), diagnosis can be confirmed by direct demonstration of *Leishmania* in tissue specimens or cultures, or by serologic testing. However, these techniques have limitations in terms of low sensitivity in general and poor performance in immunocompromised patients, respectively. Further diagnostic options have arisen, such as rapid diagnostic kits and polymerase chain reaction (PCR) tests. In general, the use of multiple diagnostic approaches is recommended to increase the likelihood of a positive result [15,46].

### 3.1. Direct Visualization of the Amastigote

Amastigotes, which are round or oval bodies 1–4 μm in diameter with a typical rod-shaped kinetoplast or circular nucleus, can be detected by direct microscopic observation with a sensitivity that depends on the collected tissue: above 90% for the spleen, 50–80% for bone marrow, and lower values for lymph node aspirates [7,15,46,47]. Blood samples have low sensitivity, except for HIV patients who exhibit higher parasitemia [7]. Tissue aspirates or biopsy specimens for smears, histopathology, parasite culture, and molecular testing are recommended; in general, bone marrow aspiration is the preferred first source of a diagnostic sample [15]. However, the need for invasive procedures to obtain a tissue specimen is an important limitation of microscopic examination for VL diagnosis [46]. Spleen aspirates, which are considered the gold standard, have an incidence of hemorrhage of up to 1/1000 procedures [48] and are routinely performed only in eastern Africa and in the Indian subcontinent [47]; bone marrow aspiration is more commonly done in Europe, Brazil, and in the United States [47].

### 3.2. Culture

Parasitological culture increases sensitivity on top of microscopy, but it is only performed in selected laboratories, and it generally leads to a diagnostic delay [47]. The microculture of noninvasive samples (buffy coat or peripheral blood mononuclear cells) seems to have a good sensitivity and the results are available in a few days up to 2 weeks [47,49,50]. In the United States, clinicians are recommended to contact their leishmaniasis reference laboratory before collecting specimens to attempt parasite isolation [15].

### 3.3. Serological Assays and Rapid Diagnostic Tests (RDTs)

Several serological assays are available, including the enzyme-linked immunosorbent assay (ELISA), the indirect fluorescent antibody test (IFAT), the indirect hemagglutination assay (IHA), immunofluorescence, and western blot. In general, these procedures show reasonable sensitivity and specificity (both 80–100% depending on test type and host factors) [15,48]. 

However, these techniques are not specific to the VL disease stage, because antibodies decrease slowly after the infection, and they are also present in asymptomatic infected patients [7]. Moreover, they cannot be used to assess response to treatment or diagnose relapses, or in the immunocompromised host [15,51]. Guidelines suggest the use of serologic testing for patients with suspected VL in whom other tests (microscopic visualization, culture, molecular tests) cannot be conducted or have negative results [15].

The development of RDTs was a step forward for the diagnosis of VL, as they are cost-effective and fast [21,47]. rK39-RDT is an immunochromatographic test which qualitatively detects antibodies which are specific for the recombinant *Leishmania* antigen rK39, a part of the kinesin-related protein of *Leishmania chagasi* [46]. This test is easy to perform and cheap and can be used for the early diagnosis of VL [51]. Its performance is considered to be high, but it varies depending on geographical areas: a Cochrane review concluded that the sensitivity of this test is excellent in the Indian subcontinent (97%), but lower in east Africa (85%) [52]. The more recently developed rK28 antigen based RDT showed a higher sensitivity in Sudan [53]. However, RDTs have the same limitations as the other serological assays and their results should be evaluated in the clinical context.

The direct agglutination test (DAT) is another serologic test that was developed to be used in areas which are endemic for VL but have limited laboratory infrastructures. It uses whole *Leishmania* promastigotes as antigens and it can be falsely positive in case of Chagas disease, brucellosis, and malaria [15,47]. A metanalysis conducted in 2006 found that DAT had a sensitivity and a specificity of 95% and 86%, respectively [54].

### 3.4. Polymerase Chain Reaction (PCR) Tests

Molecular tests can be performed on peripheral blood, bone marrow aspirates, and buffy coat samples with high sensitivity (>95%) and are currently part of the diagnostic work-up in Europe and North America [15,51]. However, some well-designed studies found low specificity, indicating that there is a risk of missing true cases, and that in endemic areas several people with asymptomatic infection are PCR positive [7,47,55]. Sensitivity was improved to 83% with loop-mediated isothermal amplification (LAMP) assay [56]. PCR has a role in the diagnosis of VL in *L. infantum*-affected countries and travel clinics, but it is rarely used in resource-limited settings mainly because of its costs.

### 3.5. Other Tests

The latex agglutination test KAtex (KALON biological, UK) detects a heat-stable low molecular weight carbohydrate antigen in urine. Its specificity is high (93%), but sensitivity is low (64%), thus, it is rarely used in clinical practice [7,47]. More recent urine antigen tests, based on ELISA techniques, show better sensitivity [57,58].

### 3.6. Diagnostic Approach

There is not a suggested diagnostic approach specific to the pediatric age. In general, in the suspicion of VL, workup should take into consideration local epidemiology and the immunocompetency status of the patient [47]. A stepwise approach should be preferred, particularly for children: first, molecular and serologic tests and microscopy should be performed on peripheral blood; if these are not sufficient to confirm or rule out VL diagnosis, tissue samples such as bone marrow or lymph nodes should be collected for further examination. 

## 4. Therapy

Treatment of VL is still very difficult and not satisfactory; chemotherapy remains the only option, with increasing drug resistances. Drugs available for this use are limited to pentavalent antimonial compounds like sodium stibogluconate (SSG) and meglumine antimoniate (MA), injectable paromomycin (PM), oral miltefosine (MF), and amphotericin B (AmpB) in two formulations (free deoxycholate, now in disuse, and lipid formulation) [59]. There are no trials investigating a therapeutic approach specific for the pediatric age: all major trials included pediatric patients and applied the same approach as for adults, or children themselves have been the main subjects of these studies [60].

Pentavalent antimonials, such as SSG, are parenteral drugs that are given in doses of 20 mg/kg for 28–30 days when used as monotherapy; their mechanism of action is still poorly understood [61]. Despite the need for prolonged parenteral treatment and the risk of adverse affects, including cardiotoxicity (ventricular tachycardia, prolonged QTc interval, ventricular fibrillation, torsades de pointe), pancreatitis, pancytopenia, and nephrotoxicity, since their discovery in 1923, these drugs have been used for decades for the treatment of VL in the vast majority of endemic regions [7,62,63]. This probably happened because of the affordability and the time-tested effectiveness of this drug [64]. 

PM, an aminoglycoside antibiotic which blocks protein synthesis, was shown to be a cheap and effective parenteral drug easily administered intramuscularly with a dosage of 15 mg/kg/day, but it requires a 21-days course. Moreover, it is potentially nephrotoxic and ototoxic [51,63].

MF, the only oral medication against VL, is an alkyl phospholipid compound developed as an antineoplastic agent against breast cancer. It had high efficacy at a dose of 2–2.5 mg/kg for 28 days when initially introduced after the first antimonial-resistant cases. It induces an increase in nitric oxide production in the macrophages that kills the parasite, alters its plasma membrane composition, and damages its mitochondria. It has a long half-life and this, together with inadequate use, unfortunately led to the induction of resistance in the protozoan. It is a teratogenic compound so it cannot be used for pregnant females. Its main adverse effects are diarrhea, vomiting, and dehydration [63,65,66] 

AmpB is an antifungal drug with high affinity binding to ergosterol, the major component of the leishmanial cell membrane; it causes the formation of aqueous pores that ultimately lead to cell death [67]. AmpB deoxycholate has major side effects including nephrotoxicity, hypokalemia, infusion reactions, and myocarditis [65,66]. This is why Liposomal AmpB (LAMB), a lipid formulation, has been developed: it has reduced toxicity, targeted drug delivery, and better pharmaco-kinetics and bioavailability. LAMB, known as Ambisome, is the only AmpB preparation approved by the Food and Drug Administration (FDA). It is expensive and requires a good cold chain, and there is remarkable geographical variation regarding the total dose administered [63,65,67]

The latter problem does not affect only this drug: unfortunately, clinical trials showed that the same therapeutic protocol could not be equally effective everywhere, with big differences based on geographical area and protozoan epidemiology [59]. At present, multidrug therapy seems to be the most promising path in many regions of the world, as it allows reduction in the duration of therapy, drug doses, and, consequently, adverse effects and costs; another advantage of this approach is that it also limits the development of drug resistances [65]. 

More than 95% of VL cases in the world are concentrated in South Asia and Eastern Africa, and it is in fact in India that increased efforts were made to develop an effective therapy for *Leishmania* eradication [59]. In the Indian subcontinent, VL patients were treated with SSG since the 1920s, but in the 1970s the first cases of unresponsiveness to pentavalent antimonials occurred. Increases in the dosage and in the duration of treatment compensated for the growing resistance in this area until the 1990s, but nowadays SSG is no longer recommended in India [7,65]. In 2005, the governments of India, Nepal, and Bangladesh launched an initiative, called the Kala-Azar Elimination Programme, and adopted MF as the first-line regimen, because of its high efficacy and ease of administration [65,68]. Unfortunately, compliance was difficult due to the long duration of the treatment and its high cost, and after about 10 years the effectiveness of this drug began to decline [7]. 

LAMB constituted a breakthrough. The price of this drug was initially too high, but, after the WHO defined guidelines on the use of LAMB depending on geographical zones in 2005, a price reduction was obtained [59,69]. Moreover, in 2010, Sundar et al. demonstrated the effectiveness of a single dose of LAMB of 10 mg/kg against *L. donovani* in patients between the ages of 2 and 65, with 95.7% efficacy and a comparable cure rate with respect to the previous treatment regimen, lasting about a month. The single dose of 10 mg/kg of LAMB is currently the first option treatment regimen in the Indian Subcontinent where a cold chain is available, while the combination therapy with PM and MF is the second choice for remote areas [65,70,71]. A recent study carried out in Bangladesh confirmed the appropriateness but also the ease of use at the level of primary care [72]. Furthermore, LAMB was donated to WHO in 2012 for 10 years, and this has been fundamental for reduction of the cases [59]. 

While in South-Eastern Asia the Kala-Azar Elimination Programme represented a joint regional approach for the fight against Leishmaniosis, in Eastern Africa a shared strategy does not yet exist and VL is still a major public health problem [65,73]. The WHO road map for neglected tropical diseases 2021–2030 established among its main objectives the reduction of the case-fatality rate of VL below 1% in Eastern Africa countries, thanks to the enhancement of diagnostic and therapeutic tools and preventive resources for transmission and vector control [74]. In this area, since 2010, the WHO recommends a multidrug therapy with SSG and PM for 17 days as the first-choice pharmacological treatment. Because of its toxicity and teratogenicity, PM cannot be used for categories such as pregnant women and people older than 45 years, who are treated with multiple doses of LAMB. Unfortunately, these therapeutic protocols are not suitable to be applied outside the hospital context in the field of primary care and LAMB is not as impactful as in South-Eastern Asia, having shown poor effectiveness in several regions, with Sudan and northern Ethiopia having similarly low efficacy rates [59,73]. LAMB has also been tested in association with SSG and with MF in Kenya and Sudan, but none of these treatments reached the target of 90% definitive cure rate at the 6-month follow-up [75]. Moreover, the latter study showed that MF has a lower efficacy in patients under 12 years of age. Probably the differences observed in the Eastern African region are caused by several factors, such as different vectors, host factors, and parasite diversity [59].

As regards South America, Brazil in one of the five countries with the highest number of annual cases in the world. The current first line therapy is still based on the administration of MA (20 mg/kg/day) for 20 days, while LAMB (3 mg/kg/day for 7 days) is the second choice [59,76] However, these recommendations result from expert consensus that rely on data obtained in other geographical areas, as there were no studies carried out in South America that prove their effectiveness [76]. In 2017, the first trials comparing antimonial compound-based regimens with those involving LAMB were carried out, and now the Brazilian Ministry of Health is considering a change in the guidelines that would make LAMB the first choice, as suggested by the WHO [59,76,77]. 

In Europe, the causative agent responsible for VL is *L. infantum*, which appears to require a higher dosage of LAMB to be eradicated compared to *L. donovani* [78,79]. In 2017, the WHO suggested seven doses of 3 mg/kg/day of LAMB, for a total of 21 mg/kg, as the best treatment regimen for VL cases on this continent [51], based on the studies of Davidson et al. published in 1994 and 1996, that, to date, are considered the most relevant trials carried out about *L. infantum* treatment and which also enrolled pediatric patients [80,81]. A total dosage of 20 mg/kg was already considered adequate for the treatment of VL in immunocompetent patients in all regions of the world in 2005, beyond the different regimens of administration [69]. Also, the American Society of Infectious Diseases (IDSA) guidelines and the FDA recommend a regimen of 3 mg/kg/day on days 1–5, day 14, and then day 21 (for a total of 21 mg/kg) for VL cases imported into North America [15]. LAMB accumulates in tissues and is then slowly released: this fact could be the reason why so many different regimens of administration are effective; pharmacodynamic studies suggest that optimal levels can be achieved if the initial dose is at least 5 mg/kg [82].

However, in more recent years, also in Europe, new studies have been carried out to investigate the effectiveness of shorter treatment regimens with LAMB, especially in Greece. In 2003, Syriopoulou et al. tested the validity of two doses of 10 mg/kg/day in children administered on two consecutive days, compared to longer-lasting regimens with LAMB and antimonial therapy: they found faster resolution of signs and symptoms in patients undergoing the shorter LAMB regimen, with better cost-effectiveness [79]. The same two-day LAMB regimen was analysed by Krepis et al. in a retrospective pediatric trial in 2017 and proved to be successful [83]. To date, however, there are no studies investigating the effectiveness of a single administration of LAMB for the treatment of VL caused by *L. infantum* in the Mediterranean basin.

At present there is no vaccine approved for the prevention of *Leishmania* infections. The only promising candidate is ChAd63-KH, a third-generation vaccine encoding two antigens of *L. donovani*, KMP-11 and HASPB [84]. It was tested in patients with PKDL and proved to be safe and immunogenic, but there is need for further studies before it can be commercialized. 

## 5. Conclusions

The WHO included VL in the latest roadmap for neglected tropical diseases, setting the development of new tools for prevention, diagnosis, and treatment of this disease as targets for 2030. Sixty-five of the 75 countries where VL is endemic have been validated for possible VL elimination as a public health problem by 2030, as has already been done in India, Bangladesh, and Nepal. 

The current context highlights the need to develop more sensitive and rapid diagnostic tests for early detection, to improve access to treatment, and to find more effective drugs for East Africa. Moreover, with respect to the risk of drug resistance and the shortage of easy-to-handle antileishmanial drugs, searching for molecules with antileishmanial properties and designing a vaccine remain priorities.

New studies in the Mediterranean area are also needed to assess the effectiveness of shorter treatment regimens, which would allow reduction in the time of hospitalization and the risk of adverse reactions and side effects. Even in these areas normally not considered as endemic, clinicians, especially pediatricians, should always keep in mind VL, given the presence of recurrent outbreaks.

## Figures and Tables

**Figure 1 microorganisms-10-01887-f001:**
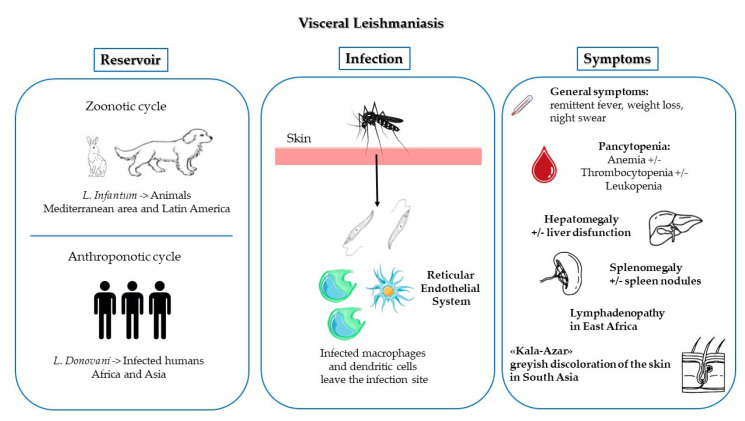
Life cycle of *L. infantum* and *L. donovani* and main symptoms of VL.

## Data Availability

Not applicable.

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
