# Peer review of "Visceral Leishmaniasis: Epidemiology, Diagnosis, and Treatment Regimens in Different Geographical Areas with a Focus on Pediatrics"

_microorganisms, 2022, doi:10.3390/microorganisms10101887_

Round 1

Reviewer 1 Report

The manuscript entitled “Visceral Leishmaniasis in children: epidemiology, diagnosis, and treatment regimens in different geographical areas” is a very informative review, well written, and easy to read. 

I have two comments , I hope the authors can address 

Chapter 1. Epidemiology, microbiology and transmission

1-    The authors did not mention the visceral leishmaniasis outbreak that occurred in the southwestern region of Madrid, Spain, which accounted for more than 494 human cases between 2009 and 2013.  I think that this epidemic is most important in Europe  

2-    In addition to the dog, other animals are thought to play a role in the transmission of L. infantum.  Hare (Lepus granatensis) and wild rabbit (Oryctolagus cuniculus) have been implicated in the Madrid epidemic (Molina R,  et al (2012) & Jiménez M, et al (2014). 

Author Response

The manuscript entitled “Visceral Leishmaniasis in children: epidemiology, diagnosis, and treatment regimens in different geographical areas” is a very informative review, well written, and easy to read. 

Authors’ reply: thank you for appreciating our work.

I have two comments , I hope the authors can address 

Chapter 1. Epidemiology, microbiology and transmission

1-    The authors did not mention the visceral leishmaniasis outbreak that occurred in the southwestern region of Madrid, Spain, which accounted for more than 494 human cases between 2009 and 2013.  I think that this epidemic is most important in Europe  

Authors’ reply: Thank you for pointing that out. We incorporated the description of the outbreak into the text (page 2).

2-    In addition to the dog, other animals are thought to play a role in the transmission of L. infantum.  Hare (Lepus granatensis) and wild rabbit (Oryctolagus cuniculus) have been implicated in the Madrid epidemic (Molina R,  et al (2012) & Jiménez M, et al (2014). 

Authors’ reply: Thank you for drawing attention to that point, we have integrated such information (page 3).

Reviewer 2 Report

This review article on Visceral Leishmaniasis covers epidemiology, clinical manifestation, disease complications, diagnosis, and therapy. The overall structure and content of the article are strong and informative, respectively. The quality of the article can be improved by addition of figures (disease manifestation and others) and tables such as chronology of epidemics in different geographical regions, current drug options and their side-effects, etc.

There are some typos and grammatical errors. There should be a coma not a period to represent 500,000 or 50,000 and others.

Title includes “Visceral Leishmaniasis in children”; however, information regarding this disease in children is limited.

Author Response

This review article on Visceral Leishmaniasis covers epidemiology, clinical manifestation, disease complications, diagnosis, and therapy. The overall structure and content of the article are strong and informative, respectively.

Authors’ reply: thank you for appreciating our work.

The quality of the article can be improved by addition of figures (disease manifestation and others) and tables such as chronology of epidemics in different geographical regions, current drug options and their side-effects, etc.

Authors’ reply: thanks for the suggestion, we followed the advice and inserted a descriptive figure of the protozoa life cycles and of the main symptoms of VL.

There are some typos and grammatical errors. There should be a coma not a period to represent 500,000 or 50,000 and others.

Authors’ reply: Thank you for spotting these inaccuracies, that we amended.

Title includes “Visceral Leishmaniasis in children”; however, information regarding this disease in children is limited.

Authors’ reply: we agree with your observations; we have changed the title into “Visceral Leishmaniasis: epidemiology, diagnosis, and treatment regimens in different geographical areas with a focus on pediatrics”.

Reviewer 3 Report

The review entitled "Visceral Leishmaniasis in children: epidemiology, diagnosis and

treatment regimens in different geographic areas" addressed the various aspects of the disease, drawing attention to the state of the art of each topic with up-to-date citations. The authors describe for each topic the need for further studies of each aspect of the disease and the attention redoubled in regions where the incidence is still very high. They also consider the lack of public health policies to improve this situation that affects hundreds of thousands of people in the world, and that in certain regions the situation is even more serious, due to several factors that are described.

In particular, in Africa the situation is alarming with the still high association of HIV circulation, which makes the clinical picture devastating. The article, in its conclusions, draws attention to the urgent need for tools to improve the prevention, diagnosis and treatment of VL and the development of drugs that have a broad spectrum of action, less resistance and short treatment as one of the priority actions, given the geographic breadth of the disease. The article serves as a guide for the deepening of the theme.

Author Response

The review entitled "Visceral Leishmaniasis in children: epidemiology, diagnosis and treatment regimens in different geographic areas" addressed the various aspects of the disease, drawing attention to the state of the art of each topic with up-to-date citations. The authors describe for each topic the need for further studies of each aspect of the disease and the attention redoubled in regions where the incidence is still very high. They also consider the lack of public health policies to improve this situation that affects hundreds of thousands of people in the world, and that in certain regions the situation is even more serious, due to several factors that are described.

In particular, in Africa the situation is alarming with the still high association of HIV circulation, which makes the clinical picture devastating. The article, in its conclusions, draws attention to the urgent need for tools to improve the prevention, diagnosis and treatment of VL and the development of drugs that have a broad spectrum of action, less resistance and short treatment as one of the priority actions, given the geographic breadth of the disease. The article serves as a guide for the deepening of the theme.

Authors’ reply: thank you for appreciating our work